# Changes in Quality and Metabolites of Pickled Purple Radish During Storage

**DOI:** 10.3390/foods14071259

**Published:** 2025-04-03

**Authors:** Seung-Hun Chae, Sang-Hyeon Lee, Seung-Hwan Kim, Si-Hun Song, Jae-Hak Moon, Heon-Woong Kim, Jeong-Yong Cho

**Affiliations:** 1Department of Horticulture and Interdisciplinary Program in IT-Bio Convergence System, Chonnam National University, Gwangju 61186, Republic of Korea; kjcmc0921@naver.com (S.-H.C.); pear@jnu.ac.kr (S.-H.L.); 2K&P Food Co., Ltd., Gyeongsan 38577, Republic of Korea; foksh2@kyochon.com; 3Department of Integrative Food, Bioscience and Biotechnology, Graduate School of Chonnam National University, Gwangju 61186, Republic of Korea; songsihun123@naver.com (S.-H.S.); nutrmoon@jnu.ac.kr (J.-H.M.); 4Department of Agronomy-Food Resources, Food and Nutrition Division, National Institute of Agricultural Sciences, Wanju-gun 55365, Republic of Korea; ksharrier@korea.kr

**Keywords:** *Raphanus sativus*, pickled purple radish, anthocyanin, metabolites, LC-QToF-MS

## Abstract

This study investigated the changes in the physicochemical properties and metabolites of pickled purple radish during storage. Pickles of purple radish (‘Boraking’) prepared by the addition of acetic acid and sugar were stored in the dark at 4 °C for 60 days. The color of the pickled purple radish changed from purple to pink, while the pickling solution changed from pink to purple. During storage, sucrose content gradually decreased, while glucose and fructose levels increased. LC-ESI-QToF-MS metabolomic analysis indicated that metabolites, including organic acids, amino acids, sulfur-containing compounds, lysophosphatidylcholine, lysophosphatidylethanolamine, and anthocyanins, were identified. The antioxidant capacity and color meter of pickled purple radish may undergo changes due to the altered levels of non-volatile compounds (cyanidins, adenosine, and amino acids) during storage. Anthocyanins had negative correlations with the color of pickled purple radish. The radical scavenging activity and ferric-reducing antioxidant power of pickled purple radish declined during storage. These findings emphasized the need for further research to develop processing and storage methods that enhance the bioactivity and stability of pickled purple radish.

## 1. Introduction

Radish (*Raphanus sativus* L., 2n = 18) is a cool season root vegetable belonging to the Brassicaceae family. It is cultivated across various regions, including Europe, America, and East Asia (China and Korea) [1,2]. In the Americas and Europe, radish is a widely cultivated short-term round type root with various skin colors (green, black, red, and purple). In contrast, in Asia, radish is a cylindrical type root cultivated year-round, and is highly disease resistant with a white skin color [3]. While the breeding of radishes in Korea has focused to improved resistance to diseases and pests, recent breeding programs have increasingly emphasized the diversification of radish cultivars in terms of both root shape—including cylindrical, round, conical, and bell-shaped forms—and skin color, such as green, red, black, and purple, in response to evolving consumer preferences [4,5,6,7].

Recently, the cultivation and consumption of colored radishes have been increasing due to growing consumers’ interest in healthy functional compounds such as flavonoids, carotenoids, and anthocyanins. Depending on pigment type, content, and structure of anthocyanin, the skin and flesh of color radishes can be white, red, black, and purple [6]. Red and purple radishes have been widely used in food, cosmetics, and as pigments due to their outstanding stability and intense coloration [6,8]. Purple radishes have an oblong root with a dark purple exterior, along with white and light purple xylem. Various purple radishes, including ‘Sweet baby’, ‘Boranam’, ‘Bordeaux’, and ‘Bora king’, have been bred and cultivated in Korea.

Liquid chromatography–mass spectrometry (LC–MS) analysis of radishes has identified 3 phenolic acids, 34 flavonoids, and 15 glucosinolates, including coumaric acid, kaempferol, quercetin, isorhamnetin, naringenin, luteolin, glucoraphanin, and glucoraphasatin [9,10,11,12,13]. Dark skin and light red skin radishes contain high contents of kaempferol, apigenin, and their derivatives, while red and purple radishes are abundant in anthocyanin (cyanidin, pelargonidin, and their derivatives) [6,14]. Additionally, the composition of these compounds differ between the skin and flesh of the radish. These compounds exhibit various biological activities, including anticancer, antifungal, anti-inflammatory, anti-obesity, antioxidant, and gastroprotective properties. Notably, anthocyanins in purple radish exhibit greater stability than in other plants, enhancing consumer preference and food quality [15,16]. Due to its coloration and the outstanding health functional effects of anthocyanin, purple radishes are used in various processed foods such as salads, pickles, and kimchi (kkakdugi). However, despite their popularity, processed products containing purple radish face challenges in developing processing and storage methods that maintain functional compound stability and overall quality.

Korean radish kimchi (kkakdugi) is one of the most widely consumed fermented foods in Korea. However, in recent years, the consumption of radish kimchi has declined, while the demand for pickled radishes has increased due to changing dietary habits [17]. Pickled radishes are manufactured with salt, vinegar, and sugar, resulting in a distinct strong sour taste and aroma. During storage, pickled radish products have faced problems of quality degradation, softening of radish tissue, and discoloration due to pH fluctuations and the high concentrations of salt and sugar in these products [18,19]. Moreover, flavonoids and phenolic acids are highly sensitive to storage conditions and processing methods [20,21]. Phenolic acids, in particular, undergo metabolite change during the pickling process, which may further alter the product’s functional properties [22]. Previous studies have investigated the use of additives to enhance metabolite stability, shelf life, and quality preservation in pickled white radish [23,24,25]. However, investigations on the qualitative properties of pickled purple radish during storage are limited. In particular, anthocyanins in purple radishes are well-known to be unstable due to several factors such as light, heat, pH, and free sugar (sucrose, fructose, and glucose) content [26,27,28]. It is considered that the degradation of anthocyanin during storage is the most important factor affecting the quality of pickled purple radish.

To address these challenges, this study aims to understand the changes in the physicochemical and antioxidant capacity of pickled purple radishes during the storage. Additionally, LC–MS-based nontargeted metabolomic analysis will be utilized to examine changes in metabolites, providing valuable insights into bioactive compounds during the storage.

## 2. Materials and Methods

### 2.1. Sample Preparation

Purple radish (‘Boraking’) was planted in December 2022 in a greenhouse (18–22 °C) located in Gokseong (Jeonnam, Korea) and harvested in February 2023. Purple radishes that were suitable for peeling and processing had to meet the following conditions: weight of more than 2 kg, length of more than 25 cm, and diameter more than 10 cm. The selected purple radish taproots were peeled and cut into 1.7 cm cubes using a specialized cutter. The purple radish cubes were immersed and mixed in a pickling solution (pH 4.0) consisting of water, acetic acid, and sugar. The purple radishes and pickling solution were packed in a polypropylene container, sealed with polypropylene film, and stored in the dark for 60 days at 4 °C. Every 15 days, pickling solution aliquots were obtained, and solid samples of the pickled purple radishes were collected and freeze-dried for further analysis.

### 2.2. Determination of the Hunter Color, Hardness, and pH

To estimate the physicochemical change and market quality of pickled purple radishes stored for different period, the following physicochemical properties were measured using a modified method from a previous study: Hunter color [1], texture (hardness) [29], and acidity (pH) [30]. The Hunter color of pickled purple radish and the pickling solution was determined by measuring the light–dark (L*), red–green (a*), and yellow–blue (b*) values using a colorimeter (NR60CP, Shenzhen 3nh Technology Co., Ltd., Shenzhen, China). The hue angle of the pickled purple radish and pickling solution was calculated using L*, a*, and b* values. Texture analysis of the pickles was performed using a hardness tester (500 N, Zwick GmbH & Co. KG, Ulm, Germany) with the following settings: test speed 5 mm/s, pretest speed 5 mm/s, and strain 75%. To confirm the storage stability of the pickling solution, the pH was measured using a pH meter (Orion star A111, Thermo Fisher Scientific, Seoul, Republic of Korea).

### 2.3. Determination of Total Phenolic (TPC) and Total Flavonoid (TFC) Contents

The total phenolic content (TPC) and total flavonoid content (TFC) were determined using a modified method from a previous study [31]. To determine the TPC of pickled purple radish, samples (150 mg dry weight [DW]) were extracted with 1 mL of 70% methanol and heated for 10 min at l00 °C using a heating block. The samples were then centrifuged (VARISPIN 12 R, CYRSTE, Gyeonggi, Republic of Korea) at 293× *g* for 10 min at 4 °C. Aliquots (10 µL) of the extract supernatants or pickling solution were reacted with 100 µL of 0.2 M Folin-Ciocâlteu solution for 3 min at room temperature. Next, each reaction solution was combined with 100 µL of 7.5% Na_2_CO_3_ and incubated for 30 min at room temperature. Finally, the TPC was determined by measuring the solution absorbance at 715 nm using a spectrophotometer (MQX200R, BioTek, Winooski, VT, USA). The TPC was expressed as the gallic acid equivalent (GAE) concentration (mg/100 g or 100 mL).

To determine the TFC of pickled purple radish, samples (150 mg DW) were extracted with 1 mL of 70% methanol and heated for 10 min at 100 °C using a heating block. The samples were then centrifuged at 293× *g* for 10 min at 4 °C. Aliquots (20 µL) of the extract supernatants or pickling solution were reacted with 180 µL of diethylene glycol for 3 min. Next, the reaction solutions were combined with 10 µL of 1 N NaOH and incubated at 37 °C for 90 min. Finally, the TFC was determined by measuring the solution absorbance at 427 nm using a spectrophotometer. The TFC was expressed as the naringin equivalent (NAE) concentration (mg/100 g or 100 mL).

### 2.4. Determination of Total Anthocyanin Content (TAC)

The total anthocyanin content (TAC) was determined using the modified method of a previous study [31]. To determine the total anthocyanin content of pickled purple radish, samples (300 mg DW) were extracted with 3 mL of 70% methanol including 2% HCl. After sonication for 30 min, the extraction solution was centrifuged at 293× *g* for 10 min at 4 °C. Aliquots (100 µL) of the pickled purple radish extract supernatants or pickling solution were analyzed by measuring the absorbance at 530 nm using a spectrophotometer. The total anthocyanin content was expressed as the cyanidin-3-glycoside (C3G) equivalent concentration (mg/100 g or 100 mL).

### 2.5. Determination of Free Sugar Content Using Gas Chromatography–Mass Spectrometry (GC–MS)

The free sugar contents were determined by GC–MS using a modified method from a previous study [32]. Pickled purple radish samples (50 mg) were extracted with 1.4 mL of methanol that was prechilled in the freezer. The extraction solution was centrifuged at 293× *g* for 3 min at 4 °C. The supernatant (700 µL) was then vortexed with 700 µL of H_2_O that was prechilled in the freezer. Next, the mixture was centrifuged at 293× *g* for 3 min at 4 °C. The concentrated supernatant (50 µL) was incubated for 90 min at 37 °C with 50 µL of freshly prepared methoxyamine in pyridine. After adding this solution to 80 µL of N-methyl-N-(trimethylsilyl)trifluoroacetamide (MSTFA) with 1% trifluoroacetamide (TMCS), the mixture was incubated for 20 min at 50 °C. The free sugar content analysis used the following GC temperature program: after holding the initial temperature of 80 °C for 2 min, the oven temperature was increased by 15 °C/min to 330 °C and then held for 5 min. The injector and detector temperatures were set at 205 and 250 °C, respectively. The sample was injected at a split ratio of 70:1. The carrier gas (helium) was maintained at a constant flow rate of 1.2 mL/min. The mass spectrometer was operated in positive electron impact mode at an ionization energy of 70.0 eV and a scan range of *m*/*z* 40–500. The free sugar content was quantified using external standards (sucrose, fructose, and glucose).

### 2.6. Analysis of Non-Volatile Metabolites by Liquid Chromatography Quadrupole Time-of-Flight Mass Spectrometry (LC-QToF-MS)

Pickled purple radish powder (150 mg DW) was extracted with 1 mL of 70% methanol overnight at room temperature under dark conditions. Afterward, the extraction solution was centrifuged at 293× *g* for 10 min at 4 °C. The supernatants were partitioned with water-saturated *n*-butanol (BuOH) in a 1:1 *v*/*v* ratio at room temperature. The BuOH layer was filtered using a 0.22 µm nylon syringe filter. Finally, 1 µL aliquots of the BuOH layer (1 mg/mL) were analyzed using an ultraperformance liquid chromatography electrospray ionization hybrid ion trap/time-of-flight mass spectrometer (Acquity UPLC™ System, Waters, Milford, MA, USA; Xevo G2-XS QToF, Waters MS Technologies, Manchester, UK). The samples were analyzed with an ACQUITY UPLC ^®^HSS T3 system (2.1 × 100 mm, 1.8 µm). Deionized distilled water including 0.1% formic acid (mobile phase A) and 100% acetonitrile including 0.1% formic acid (mobile phase B) were used at a flow rate of 0.4 mL/min with the following gradient elutions: 0 min, 2% B; 4 min, 2% B; 10 min, 10% B; 14 min, 50% B; 24 min, 90% B; 27 min, 90% B; 28 min, 2% B; and 29 min, 2% B. The column oven temperature was 40 °C. Metabolite detection was performed using a photodiode array (PDA), measuring absorbances within the wavelength range of 210–400 nm.

The MS settings were as follows: the ionization mode was ESI (positive, negative), scan range was *m*/*z* 50–1500, scan time was 0.2 sec, capillary voltage was 2.5 kV, sampling con voltage was 40 V, con gas flow was 50 L/h, desolvation gas flow was 600 L/h, desolvation temperature was 200 °C, ion source temperature was 130 °C, and collision energies were 6 eV (low) and 25~50 eV (high). Quality control (QC) analysis was performed by mixing the same amount during storage. Qualitative metabolite identification was performed using PubChem and ChemSpider.

### 2.7. Determination of the ABTS^+^ Radical-Scavenging Activity and Ferric Reducing Antioxidant Power (FRAP)

The antioxidant activity of pickled purple radish was assessed by ABTS radical (ABTS^+^) activity and ferric reducing antioxidant power (FRAP) assays using a modified method from a previous study [33]. An ABTS^+^ stock solution was prepared by reacting a 1:1 *v*/*v* mixture of 7.4 mM ABTS solution and 2.6 mM potassium persulfate solution for 12 h at room temperature in the dark. The ABTS^+^ solution was diluted with methanol to an absorbance (optical density) of 1 at 735 nm as determined using a spectrophotometer. Aliquots (10 µL) of pickled purple radish (150 mg/mL) or the pickling solution were reacted with 190 µL of ABTS^+^ solution for 2 h in the dark. The solution absorbance was measured at 735 nm. The ABTS^+^ radical-scavenging activity was expressed as the ascorbic acid equivalent (AAE) concentration (mg/100 g or 100 mL).

The FRAP solution was prepared by reacting a 10:1:1 v/v/v mixture of 300 mM acetate buffer (3.1 g of C_2_H_3_NaO_2_·3H_2_O and 16 mL of glacial acetic acid), 10 mM 2,4,6-tri(2-pyridyl)-S-triazine (TPTZ) solution, and FeCl_3_·6H_2_O solution at 37 °C. Aliquots (10 µL) of pickled purple radish (150 mg/mL) or the pickling solution were reacted with 300 µL of FRAP solution for 5 min at 37 °C in the dark. The solution absorbance was measured at 593 nm. The FRAP was expressed as the AAE concentration (mg/100 g or 100 mL).

### 2.8. Statistical Analysis

Statistical analysis of the measurements including the correlation of color (hue angle) and content of individual cyanidins was performed using GraphPad Prism 5 (Northside, San Diego, CA, USA) with Tukey’s honest significant difference (HSD) test (*p* < 0.05). Metabolites were identified using Progenesis QI [34]. One-way analysis of variance (ANOVA) and partial least squares discriminant analysis (PLS-DA) were performed with MetaboAnalyst 6.0 [35,36] using autoscaling for feature normalization.

## 3. Results and Discussion

### 3.1. Changes in the Physicochemical Properties (Color, Hardness, and pH) of Pickled Purple Radish During Storage

The color, hardness, and pH of pickled radish and pickled solution were correlated with their overall acceptability, including increased turbidity, color deterioration, sensory quality, texture softening, and microbial safety [23,37,38,39]. Therefore, we evaluated the physicochemical properties (color, hardness, and pH) of pickled purple radish and its pickling solution. Overall, the L* values of the samples did not change significantly during storage (Figure 1A). The a* values of pickled purple radish were significantly different between days 0 (16.57 ± 2.73) and 60 (12.86 ± 1.66) (Figure 1B). However, the a* values of the pickling solution were not significantly different between days 0 (0.56 ± 0.28) and 60 (0.31 ± 0.16) (Figure 1B). The b* values of pickled purple radish increased significantly during storage (Figure 1C), whereas those of the pickling solution decreased significantly. The b* values of the pickling solution were significantly higher than those of the pickled purple radish on day 30. However, by day 60, the b* values of pickled purple radish were significantly higher than those of the pickling solution. The hue angle of pickled purple radish increased significantly, whereas that of the pickling solution decreased significantly (Figure 1D). As a result, the color of pickled purple radish changed from purple to pink, while the pickling solution changed from pink to purple.

In colored crops, changes in color parameters are highly correlated with flavonoid, phenolic acid, and anthocyanin content [40,41]. During storage at 5 °C, the anthocyanin cyanidin-3,5-diglucoside in pickled perilla was reported to decrease due to its low stability, a trend similar to the color changes observed in pickled purple radish [42]. Therefore, the observed color changes in pickled purple radish can be attributed to the reduction of phenolic acids, flavonoids, and anthocyanins during storage. Kim et al. [17] reported that the moisture in pickled radish was exchanged due to osmotic pressure differences between pickled radish and the pickling solution during storage. The change in color between the pickled purple radish and the pickling solution may be caused by the exchange of water-soluble compounds, such as anthocyanin, with moisture due to osmotic pressure differences.

In the case of hardness, day 0 (40.02 ± 6.41) was significantly higher than day 15 (Figure 1E). From 15 days to 60 days of storage, the hardness of pickled purple radish did not change significantly. The hardness and texture of radish are influenced by pectin content in the cell wall, which is broken down by pectinase and polygalacturonase under conditions such as high temperatures (60–80 °C), prolonged storage, and low pH (4–5), leading to the softening of the radish flesh [2,43]. During storage at low temperature (4 °C), the hardness of pickled purple radish would have minimized relatively slowly by the low activity of pectinase and polygalacturonase.

The pH value of the pickling solution did not change significantly during storage (Figure 1F). The pH value of pickled cucumber and lotus root with beet extract was significantly changed by fermentation and osmotic pressure during the storage period [44,45]. In the pickle, the change of pH reduced the stability of organic acid, free amino acid, volatile organic compound, and water-soluble metabolites [46,47]. Therefore, in this study, the metabolites of pickled purple radish were expected to remain relatively stable due to the uniform pH of the pickling solution.

### 3.2. Changes in the TPC, TFC, and TAC of Pickled Purple Radish During Storage

Radishes have been shown to contain 609 secondary metabolites including flavonoids (38%), phenolic acids, lignans, terpenes (8.2%), glucosinolates (5.6%), terpenes, carotenoids, and anthocyanins [6,9,48,49,50]. Among them, flavonoids and phenolic acids in food are known to undergo a decrease due to processing and storage conditions (temperature, light, and CO_2_ concentration) [51,52]. In this study, we measured the total flavonoid content (TFC) and total phenolic content (TPC) of pickled purple radish to evaluate changes in functional compounds over different storage periods (Figure 2). The total phenolic content (TPC) of pickled purple radish was significantly different between 30 days (111.52 ± 2.44 mg GAE/100 g) and 45 days (95.51 ± 8.16 mg GAE/100 g) of storage. For the pickling solution, the TPC at 45 days (12.06 ± 0.67 mg GAE/100 mL) was significantly different from that of other days. The TFC of the pickling solution did not change significantly during storage (Figure 2B). White and red radishes contained total phenol content (1221–3709 mg GAE/100 g) and total flavonoid content (478–6158 mg CE/100 g) [53] which were higher than that of pickled purple radish in this experiment. The reduction in TPC and TFC during the pickling process can be attributed to the removal of the radish peel, which contains high levels of anthocyanins and phenolic compounds. However, pickled purple radish exhibited the lowest reduction in total phenolic and flavonoid content compared to various processed products [52]. Polyphenols, including catechin and anthocyanin, were unstable at high pH (>6) and their structural stability was decreased due to the structural rearrangements at high pH value [54]. Since the pickling solution maintained a uniform pH below 4, the phenolic and flavonoid contents in pickled purple radish were expected to be relatively stable. The levels of phenolic compounds have been shown to be decreased due to various enzymes, such as polyphenol oxidase and peroxidase, and peroxidase was highly activated at 50–80 °C and pH 6–7 [55]. Therefore, the low storage temperature (<4 °C) and acidic condition (pH < 4) in pickled purple radish inhibited these enzymes, contributing to the stability of total phenolic and flavonoid content. Therefore, pickled purple radish represents a viable method for polyphenol consumption while maintaining quality and health benefits.

Anthocyanin is the major secondary metabolite and functional compound contained in purple radish. Previous studies have demonstrated that anthocyanin content plays a crucial role in consumer acceptability and the bioactivity of processed purple radish products [16]. In this study, total anthocyanin content (TAC) was measured to evaluate the changes in anthocyanin levels in pickled purple radish during storage. The TAC of pickled purple radish decreased significantly during storage (Figure 2C). Specifically, the TAC at 0 days (5.95 ± 0.13 mg C3G/100 g) was significantly higher than that on other days. The TAC at 45 days (3.74 ± 0.08 mg C3G/100 g) was not significantly different from that at 60 days (3.86 ± 0.04 mg C3G/100 g). The TAC of the pickling solution did not change significantly during storage. Previous studies on purple beans, black beans, and cranberries have reported that anthocyanins, including cyanidin, pelargonidin, and delphinidin, are highly sensitive to various factors such as light, pH, sugar content, and temperature [56,57,58,59]. Amr et al. [60] reported that anthocyanins are stable under low-temperature (<4 °C), dark, and acidic (pH < 4) conditions in the absence of sugar. Also, the anthocyanin content decreased by 5, 30, and 80% at the three different pH values (pH 3, 4, and 5) for 210 min [60]. Despite the low pH and storage temperature, the reduction of anthocyanin in pickled purple radish will be influenced by the low stability. Anthocyanins (delphinidin-3-galactoside, cyanidin-3-galactoside, cyanidin-3-arabinoside, etc.) and total anthocyanin content in blueberries were reduced by 30–75% within 30 days of storage [61]. In this study, the reduction of anthocyanin content in pickled purple radish was approximately 25% during storage. The low reduction of anthocyanin in pickled purple radish is suggested to be due to their structural stability in pH 4.0.

### 3.3. Change in the Free Sugar (Fructose, Glucose, and Sucrose) Content of Pickled Purple Radish During Storage

Free sugar content is an important factor influencing the taste (sweetness), color, and texture of food [62]. Additionally, an increase in free sugar content can lead to the generation of reactive oxygen species (ROS), resulting in cell membrane leakage [63]. In this study, we analyzed the levels of fructose, glucose, and sucrose in pickled purple radish and the pickling solution during storage (Figure 3). Among the free sugars detected, the highest concentrations were observed in the following order: sucrose > glucose > fructose. The sucrose content in pickled purple radish decreased during storage. The sucrose content (109.22 ± 0.74 µmole/g) at 0 days was significantly higher than that (74.02 ± 9.46 µmole/g) at 60 days. However, the content of fructose and glucose increased over the storage period. In particular, the glucose content (43.07 ± 6.00 µmole/g) on day 60 was significantly higher than that (31.64 ± 2.54 µmole/g) on day 0, and the fructose content (41.70 ± 5.88 µmole/g) on day 60 was significantly higher than that (29.60 ± 2.44 µmole/g) on day 0. Glucose and fructose content in salted white radish decreased rapidly within 20 days of fermentation [64]. The free sugars in root crop such as radishes, sweet potatoes, and potatoes were also hydrolyzed during storage [65,66,67,68]. These results suggest that sucrose is hydrolyzed to glucose and fructose in pickled purple radish during storage. Free sugars have different relative sweetness levels as perceived by the human mouth [fructose (1.5–1.8) > sucrose (1.0) > glucose (0.5)] [69,70]. Despite the decrease in sucrose levels during storage, the overall sweetness of pickled purple radish remains relatively unchanged due to the increased fructose content, which has a higher relative sweetness.

### 3.4. Metabolites of Pickled Purple Radish During Storage

Metabolite analysis of pickled purple radish was conducted using LC-QToF-MS to confirm the changes in metabolites and bioactive compounds during storage. Metabolites in pickled solution were not detected, which may be their trace or small amount. Thirty-four metabolites (12 amino acids, 1 sulfur compound, 5 anthocyanins [cyanidins], 1 lignan, and 15 fatty acids) from pickled purple radish were detected in the LC-QToF-MS chromatogram (Figure 4, Table 1).

Five cyanidin derivatives (Peaks 17, 18, 19, 20, and 21) were identified based on previous studies and fragmentation at LC-QToF-ESI-MS [9,71]. Cyanidin-3-*O*-caffeoylsophoroside-5-*O*-malonyldiglucoside (Peak 17) was detected at observed *m*/*z* 1021.2479 [M]^+^, 773.1924 [M-Caf-Mal-Glu]^+^, and 287.0566 [M-Caf-Mal-4Glu]^+^. Cyanidin-3-*O*-feruloylsophoroside-5-*O*-malonylglucoside (Peak 18) was detected at observed *m*/*z* 1035.2627 [M]^+^, 787.3702 [M-Mal-Glu]^+^, and 287.1151 [M-Fer-Mal-4Glu]^+^. Cyanidin-3-*O*-coumaroylsinapoylsophoroside-5-*O*-malonylglucoside (Peak 19) was detected at observed *m*/*z* 1373.3645 [M]^+^, 963.2558 [M-Mal-Glu]^+^, and 697.1616 [M-Cou-Sin-2Glu]^+^. Cyanidin-3-diferuloylsophoroside-5-malonylglucoside (Peaks 20 and 21) was detected at observed *m*/*z* 1211.3108 [M]^+^, 1035.2664 [M-Fer]^+^, and 963.2518 [M-Mal-2Glu]^+^ (Figure 5). Numerous cyanidin and pelargonidin derivatives conjugated with caffeic acid, ferulic acid, and glycoses have been reported to be found in purple radish by LC–MS metabolomic analysis [9]. Various anthocyanins were distributed in purple radish skin compared to its flesh. However, in this study, only five cyanidin derivatives were detected in pickled purple radish, which may be due to loss of various anthocyanins during peeling.

Raphanin, 5 LysoPC, and 4 LysoPE were also identified as major metabolites in pickled purple radish. Raphanin is a volatile sulfur compound responsible for the pungent flavor of radish [72]. Glucoraphasatin and glucoraphenin are the predominant glucosinolate found in radishes and are known to be hydrolyzed into raphasatin and raphanin, depending on myrosinase and low pH conditions [73,74,75]. In colored radishes, glucoraphasatin is predominantly contained in the skin, whereas glucoraphenin is detected in the skin and flesh [76]. Various sulfur compounds except raphanin seem to be the loss during peeling and/or degradation in low pH. LysoPC and LysoPE are the major phospholipids in the cell membrane [77]. These metabolites may be derived from the cell membrane of purple radish. A previous study has reported that metabolites, such as amino acids (11.24%), flavonoids (2.75%), phenolic acids (1.83%), terpenoids (9.63%), and lipids (20.87%), were found in pickled white radish [25]. Our LC–MS metabolomic results indicated that amino acids (tyrosine, isoleucine, etc.), 5 lysophosphatidylcholine (LysoPC), and 4 lysophosphatidylethanolamine (LysoPE) with 5 anthocyanins were the main metabolites in pickled purple radish.

Figure 6A presents clustering patterns among pickled purple radish samples of different storage points using principal component analysis (PCA) score plots. Significant metabolite changes were observed after 15 days of storage, but the metabolites were relatively stable after 30 days. The 30-, 45-, and 60-day storage groups were clustered, indicating similar metabolic compositions on the PCA score plot. Based on one-way analysis of variance (ANOVA), 23 compounds, including 5 anthocyanins, were significantly different according to Tukey’s HSD (*p* < 0.05; Figure 6B). Figure 6C shows that the intensity of metabolites in pickled purple radish changed during storage. The levels of pyroglutamic acid, leucyl-leucine, and isoleucine were significantly changed during storage. A previous study observed that branched-chain amino acids (BCAAs), such as isoleucine and leucine, accumulated in salt-pickled white radish due to the suppressed myrosinase activity during storage [78]. Raphanin was the highest relative intensity in pickled purple radish samples of 15 days and then significantly decreased. Raphanin might be produced by the hydrolysis of glucoraphenin under low pH condition during the storage, but decreased by high volatility after 15 days of storage [72]. Miquelianin and adenosine among the metabolites significantly decreased levels during storage. Miquelianin is known for its antioxidant capacity and antidiabetic effect due to inhibition of α-amylase and α-glucosidase [79]. A decrease in the content of adenosine derivatives has induced softening and browning in fruits and vegetables due to the destruction of the membrane structure and function [80]. The change of adenosine is associated with the softening and browning of radish tissue, suggesting their potential as indicator compounds for evaluating the degree of hardness in pickled purple radish. Five cyanidin derivatives gradually decreased in pickled purple radish during storage. Individual cyanidin derivatives had relative intensity in the order of peak 18 > peak 20 > peak 19 > peak 17 (Figure 6D). The individual cyanidin contents and TAC in pickled purple radish showed a similar decreasing tendency, the four cyanidin contents were reduced within the first 15 days. In particular, peak 18 (cyanidin-3-*O*-feruloylsophoroside-5-*O*-malonylglucoside), having high relative intensity, was largely reduced during the storage. From these results, these compounds may reduce by a high amount through degradation during storage.

To investigate the relationship between metabolite changes and chromaticity, a correlation analysis was conducted between cyanidin content and the hue angle of pickled purple radish during storage. Four cyanidin derivatives had a negative correlation with the hue angle, showing a Pearson correlation coefficient (r) of −0.85 to −0.75 (Figure 7). Cyanidin-3-*O*-feruloylsophoroside-5-*O*-malonylglucoside had the highest coefficient of determination (R^2^ value = 0.71). Due to the decreased anthocyanin levels during storage, the chromaticity of pickled purple radish changed from purple to pink, and the hue angle decreased. These results suggested that cyanidin derivatives were an important biomarker for the color quality of pickled purple radish.

### 3.5. Changes in the ABTS^+^ Radical-Scavenging Activity and FRAP of Pickled Purple Radish During Storage

The ABTS^+^ radical scavenging activity (as measured by the ABTS^+^ assay) and ferric reducing capacity (as measured by the FRAP assay) of pickled purple radish were analyzed to confirm the changes in bioactivity of the metabolites during the storage period. The results showed that the ABTS^+^ radical scavenging activity and FRAP of pickled purple radish and the pickling solution decreased significantly during storage (Figure 8). The ABTS^+^ and FRAP values of pickled purple radish and solution were significantly decreased during storage. In a previous study, the ABTS^+^ and FRAP values of a variety of crops were influenced by the TAC, TPC, and TFC during storage [81,82]. The decreasing trend of the ABTS^+^ and FRAP values in pickled purple radish was consistent with the decline in the TAC, individual anthocyanin, and functional components (miquelianin, and adenosine). Previous studies reported that the hydroxyl, methoxy, and carboxyl groups in phenolic compounds contribute to high antioxidant activity by influencing enzymatic reactions such as deglycosylation, dehydroxylation, demethylation, and oxidation [83,84]. The hydroxyl and carboxyl groups of cyanidins contributed to its enhancing solubility and antioxidant activity. Therefore, it is essential to minimize the loss of these bioactive compounds to sustain the antioxidant capacity of pickled purple radish during storage. Also, it has been shown that these functional compounds affect not only bioactivity but also quality, such as color and browning [9,10,11,12,13,79,80].

As mentioned above, the radish peel contained twice as the number of functional components (phenolic acid and flavonoid) as the flesh. In most varieties of purple radish, a large amount of anthocyanin was concentrated in the skin rather than in the flesh [85,86]. Cyanidin, such as cyanidin-3-glucoside, had different chromaticity (L*, a*, and b*) and bioactivity (anti-cardiovascular, anticancer, antioxidant, and anti-inflammatory effects) depending on the structure and number of conjugated amino acids and glycosides [87,88]. Cyanidin derivatives are considered to be the main metabolites that affect the color and antioxidants of pickled purple radish during storage. However, storage stabilities and bioactivities of cyanidin derivatives, including cyanidin-3-*O*-diferuloylsophoroside-5-*O*-malonylglucoside and cyanidin-3-*O*-caffeoylsophoroside-5-*O*-malonyldiglucoside, have not been studied. Investigation on the bioactivities of cyanidin derivatives from purple radish and their degradation mechanism during storage would be required. In addition, studies on processing methods to minimize the loss of functional compounds (anthocyanins) by peeling and storage methods to enhance anthocyanin stability are required.

## 4. Conclusions

In this study, changes in the physicochemical properties and metabolites in pickled purple radish prepared with addition of acetic acid and sugar at 4 °C were evaluated during the storage. During storage, the color, hardness, and total anthocyanin content of the pickled purple radish gradually reduced, but the total phenolic and flavonoid content did not change. Metabolites, including cyanidin derivatives, raphanin, lysoPC, and lysoPE, were found as the main compounds in the pickled purple radish through LC-ESI-QToF-MS metabolomic analysis. In particular, cyanidin derivatives gradually decreased in the pickled purple radish during the storage, which showed a negative correlation with color. Anthocyanin is considered as an important biomarker for the bioactivity and color of pickled purple radish. However, further studies on the processing and storage methods to minimize the loss and degradation of functional compounds (anthocyanins) should be conducted. These results provide valuable information on the processing and storage methods for processed foods using purple radish.

## Figures and Tables

**Figure 1 foods-14-01259-f001:**
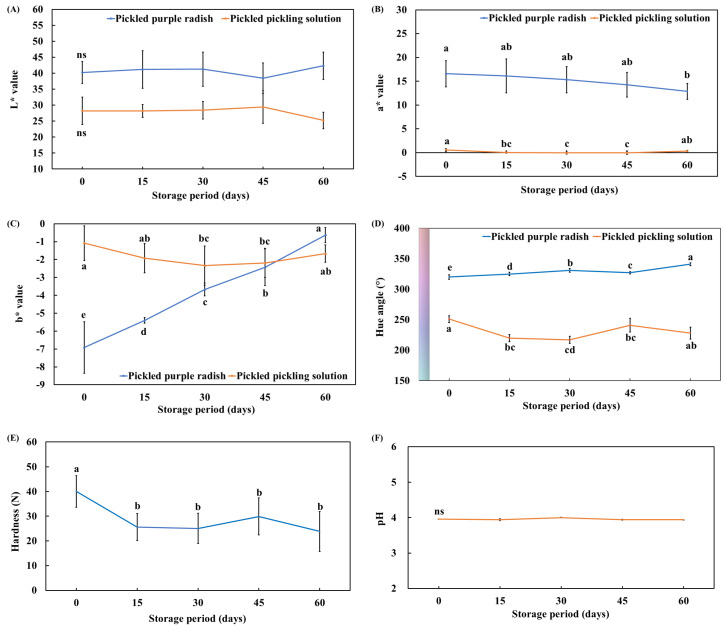
L* value (**A**), a* value (**B**), b* value (**C**), hue angle (**D**), hardness (**E**), and pH (**F**) of pickled purple radish during storage. Different letters at the lines indicate significant differences according to Tukey’s HSD test (*p* < 0.05).

**Figure 2 foods-14-01259-f002:**
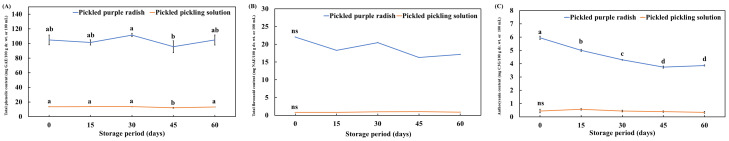
Total phenolic content (**A**), total flavonoid content (**B**), total anthocyanin content (**C**) of pickled purple radish during storage. Different letters above the bars indicate significant differences according to Tukey’s HSD test (*p* < 0.05).

**Figure 3 foods-14-01259-f003:**
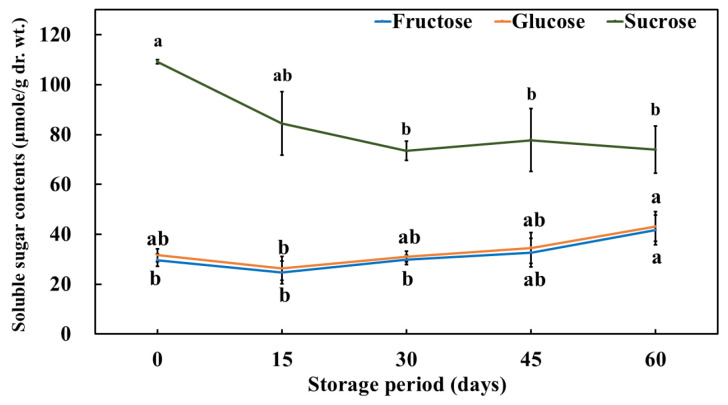
Free sugar (fructose, glucose, and sucrose) content of pickled purple radish during storage. Different letters above the bars indicate significant differences according to Tukey’s HSD test (*p* < 0.05).

**Figure 4 foods-14-01259-f004:**
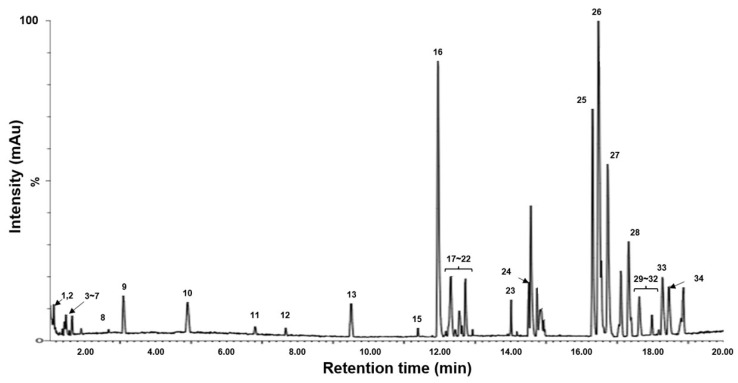
Representative LC-QToF-MS chromatogram of pickled purple radish. The 34 compounds include amino acids, organic acids, anthocyanins, lysophosphatidyl choline (LPC), and lysophosphatidyl ethanolamine (LPE).

**Figure 5 foods-14-01259-f005:**
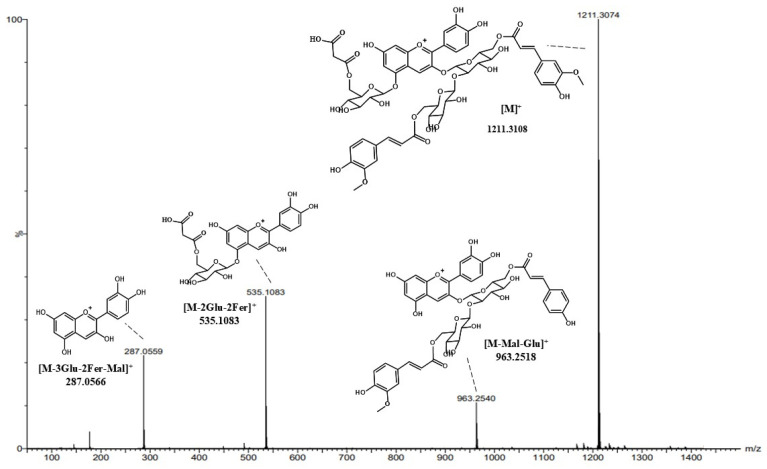
MS (positive) spectrum and fragmentation of cyanidin-3-diferulouylsophoroside-5-malonylglucoside (peak 20). Glu: glucose, Mal: malic acid, Fer: ferulic acid.

**Figure 6 foods-14-01259-f006:**
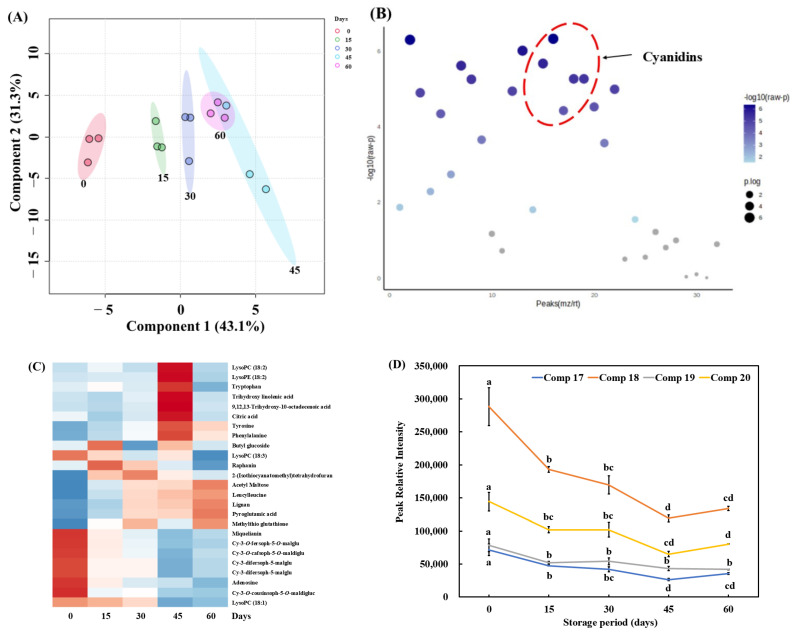
Principal component analysis (PCA) score plot (**A**), one-way analysis of variance (ANOVA) (**B**), heatmap (**C**), and change in relative intensities of main anthocyanins (**D**) in pickled purple radish during storage. Heatmap of metabolites in red and blue represent those upregulated and downregulated. Different letters above the line indicate significant differences according to Tukey’s HSD test (*p* < 0.05). Peak 17: cyanidin-3-*O*-caffeoylsophoroside-5-*O*-malonyldiglucoside, Peak 18: cyanidin-3-*O*-feruloylsophoroside-5-*O*-malonylglucoside, Peak 19: cyanidin-3-*O*-coumaroylsinapoylsophoroside-5-*O*-malonylglucoside, Peak 20: cyanidin-3-diferuloylsophoroside-5-malonylglucoside. Cy: cyanidin; Fer: ferulic acid; Soph: sophoroside; Mal: malonic acid; Glu: glucose; Caf: caffeic acid; Cou: coumaric acid, Sin: sinapinic acid.

**Figure 7 foods-14-01259-f007:**
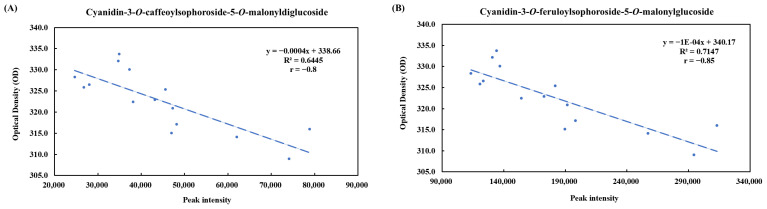
Pearson’s correlations between color (hue angle) and cyanidin-3-*O*-caffeoylsophoroside-5-*O*-malonyldiglucoside (peak 17, (**A**)), cyanidin-3-*O*-feruloylsophoroside-5-*O*-malonylglucoside (peak 18, (**B**)), cyanidin-*O*-coumaroylsinapoylsophoroside-5-*O*-malonylglucoside (peak 19, (**C**)), and cyanidin-3-*O*-diferuloylsophoroside-5-*O*-malonylglucoside (peak 20, (**D**)).

**Figure 8 foods-14-01259-f008:**
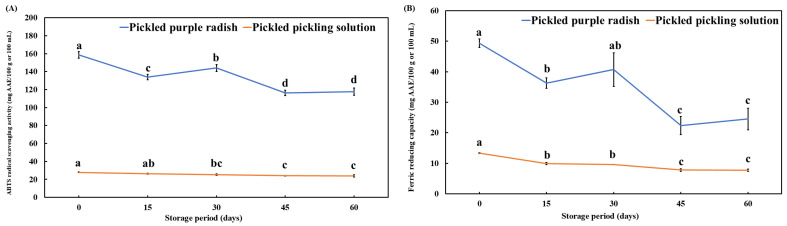
ABTS^+^ radical scavenging activity (**A**) and ferric reducing antioxidant power (**B**) of pickled purple radish and pickling solution during storage. Different letters above the bars indicate significant differences according to Tukey’s HSD test (*p* < 0.05).

**Table 1 foods-14-01259-t001:** MS results of metabolites identified in pickled purple radish.

No.	t_R_ (min)	Fragment Ions (*m*/*z*)		Mode of Ionization	Theoretical (*m*/*z*)	Observed (*m*/*z*)	Mass Error (ppm)	Molecular Formula	Predicted Compounds
		Positive	Negative						
1	1.02	68.9973	111.0090, 87.0100	[M-H]^−^	191.0212	191.0200	4.7	C_6_H_8_O_7_	Citric acid
2	1.13	84.0458	128.0311	[M+H]^+^	130.0547	130.0513	4.2	C_5_H_7_NO_3_	Pyroglutamic acid
3	1.39	141.9600	117.0195, 73.5326	[M-H]^−^	117.0188	117.0195	5.0	C_4_H_6_O_4_	Succinic acid
4	1.42	165.0558	163.0397, 118.0229	[M+H]^+^	182.0791	182.0826	4.9	C_9_H_11_NO_3_	Tyrosine
5	1.57	86.0979	-	[M+H]^+^	132.1024	132.1033	5.1	C_6_H_13_NO_2_	Isoleucine
6	1.65	205.0718, 61.0081	59.6404	[M+H]^+^	385.1368	385.1022	1.5	C_14_H_24_O_12_	Acetyl maltose
7	1.90	136.0632	134.0462	[M+H]^+^	268.1051	268.1050	1.5	C_10_H_13_N_5_O_4_	Adenosine
8	2.68	126.9580	-	[M+H]^+^	144.0483	144.0493	4.5	C_6_H_9_NOS	2-(Isothiocyanatomethyl)tetrahydrofuran
9	3.09	120.0822	103.6744	[M+H]^+^	166.0868	166.0876	4.8	C_9_H_11_NO_2_	Phenylalanine
10	4.90	308.8376	-	[M+H]^+^	354.0715	354.0798	1.1	C_11_H_20_N_3_O_6_S_2_	Methylthio glutathione
11	6.81	188.0718	159.8946	[M+H]^+^	205.0972	205.0981	3.2	C_11_H_12_N_2_O_2_	Tryptophan
12	7.67	235.9700	-	[M+Na]^+^	259.1329	259.1165	2.7	C_10_H_20_O_6_	Butyl hexose
13	9.50	235.9700		[M+Na]^+^	259.1329	259.1165	2.7	C_10_H_20_O_6_	Butyl hexose
14	10.93	-	151.9011	[M-H]^−^	477.0697	477.0643	−5.4	C_21_H_18_O_13_	Miquelianin
15	11.39	86.0975	-	[M+H]^+^	245.1884	245.1870	2.0	C_12_H_24_N_2_O_3_	Leucylleucine
16	11.95	112.0230		[M+H]^+^	176.0204	176.0219	4.5	C_6_H_9_NOS_2_	Raphanin
17	12.02	773.1924, 287.0566	-	[M]^+^	1021.2461	1021.2479	1.8	C_45_H_49_O_27_	Cyanidin-3-*O*-caffeoylsophoroside-5-*O*-malonyldiglucoside
18	12.32	787.3702, 287.1151	-	[M]^+^	1035.2618	1035.2627	0.9	C_46_H_51_O_27_	Cyanidin-3-*O*-feruloylsophoroside-5-*O*-malonylglucoside
19	12.44	963.2558, 697.1616	1371.35	[M]^+^	1373.3617	1373.3645	1.9	C_62_H_69_O_35_	Cyanidin-3-*O*-coumaroylsinapoylsophoroside-5-*O*-malonyldiglucoside
20	12.55	1035.2664, 963.2518	-	[M]^+^	1211.3093	1211.3108	1.4	C_56_H_59_O_30_	Cyanidin-3-*O*-diferuloylsophoroside-5-*O*-malonylglucoside
21	12.73	1035.2664, 963.2518	-	[M]^+^	1211.3093	1211.3108	1.4	C_56_H_59_O_30_	Cyanidin-3-*O*-diferuloylsophoroside-5-*O*-malonylglucoside isomer 1
22	12.96	-	160.0445	[M-H]^−^	693.2036	693.2034	−4.6	C_32_H_38_O_17_	Lignan
23	14.01	279.0948, 81.5212	211.1340	[M-H]^−^	327.2162	327.2179	2.4	C_18_H_34_O_5_	Trihydroxylinolenic acid
24	14.28	169.9780, 97.9925	265.1494	[M-H]^−^	329.2328	329.2334	1.8	C_18_H_34_O_5_	9,12,13-Trihydroxy-10-octadecenoic acid
25	16.31	335.2590	542.2498, 277.2166	[M+H]^+^	476.2652	476.2780	0.6	C_23_H_42_NO_7_P	LysoPE(18:3)
26	16.47	540.3062, 335.2580	552.2858, 277.2175	[M+H]^+^	518.3228	518.3246	−0.2	C_26_H_48_NO_7_P	LysoPC(18:3)
27	16.73	540.3063, 335.2588	552.2856, 277.2162	[M+H]^+^	518.3228	518.3246	−0.2	C_26_H_48_NO_7_P	LysoPC(18:3)
28	17.10	423.2740, 337.2740	279.2337	[M+H]^+^	478.2881	478.2930	−0.8	C_23_H_44_NO_7_P	LysoPE(18:2)
29	17.32	502.3282, 337.2740	554.3004	[M+H]^+^	520.3048	520.3397	−1.2	C_26_H_50_NO_7_P	LysoPC(18:2)
30	17.63	502.3281, 337.2718	554.2996	[M+H]^+^	520.3048	520.3397	−1.2	C_26_H_50_NO_7_P	LysoPC(18:2)
31	17.98	436.2665, 313.2737	255.9257	[M+H]^+^	454.2812	454.2928	−1.3	C_21_H_44_NO_7_P	LysoPE(16:0)
32	18.15	339.2892, 223.9893	281.2467	[M+H]^+^	480.3247	480.3084	−1.2	C_23_H_46_NO_7_P	LysoPE(18:1)
33	18.28	459.2485, 104.1074	480.3099	[M+H]^+^	496.3356	496.3398	−1.0	C_24_H_50_NO_7_P	LysoPC(16:0)
34	18.45	504.3404, 184.9869	566.3459, 281.2491	[M+H]^+^	522.3558	522.3554	−1.1	C_26_H_52_NO_7_P	LysoPC(18:1)

## Data Availability

The original contributions presented in the study are included in the article, further inquiries can be directed to the corresponding author.

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
