# Peer review of "Changes in Quality and Metabolites of Pickled Purple Radish During Storage"

_foods, 2025, doi:10.3390/foods14071259_

Round 1

Reviewer 1 Report

Comments and Suggestions for Authors

A well-constructed MS. Here are some comments and recommendations:

  1. Line 54: Red40 is made from petroleum. The references cited here are not suitable for this.
  2. Line 74: The ref #19 is talking about amelioration of the radish process rather than changing dietary habits. The references cited here are not suitable for this.
  3. Line122-123: Why only pH for the stability of pickling solution? Do you need to check other indicators such as the color change, sugar content and microbial load etc?
  4. Line296-299: The pH of pickling solution has pH at approx. 4 for the whole storage period. Also the reference mentioned here suggests pH>4 is required to cause unstable of polyphenols. Therefore, your data indicates TPC and TFC showing insignificant change for the 60 days storage. It seems that pH is not the major factor for polyphenols reduction as mentioned here.
  5. Line 367-370: The sweetness can be guess?? You need to provide sensory evaluation result or sweetness converting ratio for fructose vs. sucrose, and glucose vs. sucrose.

Author Response

Line 54: Red40 is made from petroleum. The references cited here are not suitable for this.
Answer: We appreciated for your valuable comment. We removed the mention of ‘Red 40’ in line 49 of the revised manuscript, as it was not considered to the main context of the paper.

Line 74: The ref #19 is talking about amelioration of the radish process rather than changing dietary habits. The references cited here are not suitable for this.
Answer: We changed the reference in line 71 of the revised manuscript, as it was suitable main context of the paper

Line122-123: Why only pH for the stability of pickling solution? Do you need to check other indicators such as the color change, sugar content and microbial load etc?
Answer: We appreciated for your valuable comment. White radish pickles are commercially produced and consumed with no safety issues (microbial contamination). Purple radish pickles were also thought to have no safety issues due to microbial contamination during storage. However, anthocyanins in purple radish pickles are well known to be very unstable and could be easily changed by pH, heat, light, etc. during storage. In this study, the purple radish pickles were deposited in the dark and low temperature (4 ℃) during storage. In particular, pH change of purple radish pickles could be influenced by the materials during storage. Therefore, we thought that pH was the most important factor for anthocyanin and polyphenolic compounds degradation as biological compounds in purple radish pickles. In this study, although individual anthocyanins may change slightly, pH and color did not change in pickle solution during storage.

Line296-299: The pH of pickling solution has pH at approx. 4 for the whole storage period. Also the reference mentioned here suggests pH>4 is required to cause unstable of polyphenols. Therefore, your data indicates TPC and TFC showing insignificant change for the 60 days storage. It seems that pH is not the major factor for polyphenols reduction as mentioned here.
Answer: We appreciated for your valuable comment. Polyphenolics are known to be commonly stable at pH < 6. We revised this with reference citation in line 285 of the revised manuscript.

Line 367-370: The sweetness can be guess?? You need to provide sensory evaluation result or sweetness converting ratio for fructose vs. sucrose, and glucose vs. sucrose.
Answer: We appreciated for your valuable comment. We described relative sweetness level with citation in line 343 of the revised manuscript. 

Reviewer 2 Report

Comments and Suggestions for Authors

Dear Editor and Authors,

I am very grateful you for the invitation to review the manuscript by Seung-Hun Chae1, and co-authors Changes in Quality and Metabolites of Pickled Purple Radish during Storage

The paper deals with the quality and metabolite changes in purple radish.

Comments on the paper are as follows.

Page 2 Line 54 The Authors write “Red and purple radishes have been widely used in food, cosmetics, and the pigment Red 40 due to their outstanding stability and intense coloration” what does red 40 mean?

Page 2 Line 61 , 62 citing literature line 61 is the same as line 62.  In the reviewer's opinion it is enough to quote once in line 62

Page 3 line 129-130 and 138-139 The authors write:

“To determine the TPC of pickled purple radish, samples (150 mg dry weight [DW]) were extracted with 1 mL of 70% methanol and heated for 10 min at l00 ℃ using a heating block.” “To determine the TFC of pickled purple radish, samples (300 mg DW) were extracted  with 1 mL of 70% methanol and heated for 10 min at l00 ℃ using a heating block.”

Question of the reviewer is whether the solvent volume of 1 ml in relation to the 150 or 300 mg /DM products weight was sufficient to extract the polyphenols . Did the authors study the recovery?

Page 4 line 145, 154 Are the results expressed in g/100 g of fresh product or g/100 g of freeze-dried product or g/100 g of DM?

Page 4 Line 174 Why did the authors obtain only aqueous solutions for  liquid chromatography quadrupole time-of-flight  mass spectrometry (LC-QToF-MS) analysis? In the opinion of the reviewer, it would be important to examine water-alcohol extracts that contain hydrophilic and hydrophobic compounds.  Aqueous solutions obtained at room temperature may not contain hydrophobic compounds

Page 7 Are the results of phenolic expressed in g/100 g of fresh product or g/100 g of freeze-dried product or g/100 g of DM?

Page 8 Line 440-442 The Authors write “In this study, amino acids (citric acid, tyrosine, isoleucine,  etc.), 5 lysophosphatidylcholine (LysoPC), and 4 lysophosphatidylethanolamine (LysoPE) were detected in pickled purple radish.” According to the reviewer, not all compounds are amino acids

Page 10  Line 369 Table 1 is not very readable

Page 13 Line 468-471 The Authors write “Four cyanidins had a negative correlation with the  hue angle, showing a coefficient of determination of 0.6 to 0.7 (Figure 7). Cyanidin-3-O- feruloylsophoroside-5-O-malonylglucoside had the highest coefficient of determination  (R2 value=0.71)” Reviewer's question: Can we say that the correlation is negative?

Author Response

Page 2 Line 54 The Authors write “Red and purple radishes have been widely used in food, cosmetics, and the pigment Red 40 due to their outstanding stability and intense coloration” what does red 40 mean?
Answer: Pigments are used as alternatives to Red 40, a commercial food colorant. We removed the mention of ‘Red 40’ in line 49 of the revised manuscript, as it was not considered to the main content of the paper.

Page 2 Line 61 , 62 citing literature line 61 is the same as line 62.  In the reviewer's opinion it is enough to quote once in line 62
Answer: We appreciated for your comment. References were presented in line 59 of the revised manuscript. 

Page 3 line 129-130 and 138-139 The authors write:
“To determine the TPC of pickled purple radish, samples (150 mg dry weight [DW]) were extracted with 1 mL of 70% methanol and heated for 10 min at l00 ℃ using a heating block.” “To determine the TFC of pickled purple radish, samples (300 mg DW) were extracted with 1 mL of 70% methanol and heated for 10 min at l00 ℃ using a heating block.”
Question of the reviewer is whether the solvent volume of 1 ml in relation to the 150 or 300 mg /DM products weight was sufficient to extract the polyphenols. Did the authors study the recovery?
Answer: We appreciated for your valuable comment. We found an important error for the sample weight of TFC and this revised 300 mg DW to 150 mg DW in line 129 of the revised manuscript. The samples (150 mg DW) were extracted with 1 mL of 70% methanol, the standard deviations (<10%) of their extraction yields were not high. In addition, we corrected 150 mg DW of sample weight for LC-MS metabolite analysis in line 165 of the revised manuscript. We would be very grateful if you agree with us.

Page 4 line 145, 154 Are the results expressed in g/100 g of fresh product or g/100 g of freeze-dried product or g/100 g of DM?
Answer: We appreciated for your comment. We inserted DW in Figures. 

Page 4 Line 174 Why did the authors obtain only aqueous solutions for liquid chromatography quadrupole time-of-flight mass spectrometry (LC-QToF-MS) analysis? In the opinion of the reviewer, it would be important to examine water-alcohol extracts that contain hydrophilic and hydrophobic compounds.  Aqueous solutions obtained at room temperature may not contain hydrophobic compounds
Answer: We agreed to your valuable comment. We analyzed non-volatile metabolites in both water extracts and 70% methanol extracts by LC-QToF-MS. The metabolites in both water extracts and 70% methanol extracts were similar as below. However, metabolites, including lysophospholipids (PE and PC) in 70% methanol extracts were higher intensities compared to those in water extracts. LC-QToF-MS chromatogram of 70% methanol extracts were presented in Figure 4. However, we miswritten deionized distilled water as extraction solvent in section 2.6 analysis of non-volatile~LC-QToF-MS). We revised 70% methanol in line 165 of the revised manuscript. We would be very grateful if you agree with us.

Page 7 Are the results of phenolic expressed in g/100 g of fresh product or g/100 g of freeze-dried product or g/100 g of DM?
Answer: We appreciated for your comment. All figures are expressed in dry weight. 

Page 8 Line 440-442 The Authors write “In this study, amino acids (citric acid, tyrosine, isoleucine, etc.), 5 lysophosphatidylcholine (LysoPC), and 4 lysophosphatidylethanolamine (LysoPE) were detected in pickled purple radish.” According to the reviewer, not all compounds are amino acids
Answer: We appreciated for your comment. Citric acid is deleted in line 393 of the revised manuscript.

Page 10 Line 369 Table 1 is not very readable
Answer: The modified Table 1 is expressed in line 364. The origin of Table 1 is attached as Word file. Please, check to the origin if the modified Table 1 is not wrong.

Page 13 Line 468-471 The Authors write “Four cyanidins had a negative correlation with the hue angle, showing a coefficient of determination of 0.6 to 0.7 (Figure 7). Cyanidin-3-O- feruloylsophoroside-5-O-malonylglucoside had the highest coefficient of determination (R2 value=0.71)” Reviewer's question: Can we say that the correlation is negative?
Answer: We understand that the coefficient of determination (R²) ranges from 0 to 1, which means to linear correlation of the data fit. Also, Pearson correlation coefficient (r) ranges from -1 to 1, which reflects the strength and direction (positive and negative) of the linear correction. we have now clearly distinguished between these two metrics and additionally included the Pearson correlation coefficients to better explain degree of correlation between individual anthocyanins and hue angle in line 441

Reviewer 3 Report

Comments and Suggestions for Authors
  1. The abstract is too long, and the abstract only needs to reflect the important results and conclusions of the study. Too many descriptive results in the abstract.
  2. In the introduction, the author only introduces Korean radishes, which is one-sided because radishes are grown and processed in many countries around the world. What's the difference from the others? What are the advantages?
  3. What are the differences between different varieties, even different colors and places of origin, and the components analyzed by LC-MS? The author should go into detail in the introduction.
  4. The innovation of the research is not introduced and mentioned in the introduction.
  5. Among the materials, the radish is from December 2022 to February 2023. And the current opinion over the past two years, whether the processed materials still meet the food safety?Authors should provide data, such as microbes.
  6. 1 and Fig. 2 Because the combined picture is too small, it is difficult to see the content in the picture. Please modify and enlarge the picture and content.
  7. Section 3.5. In this part, the author only describes the results and phenomena, and the discussion on the results is insufficient.
  8. The author should improve the clarity of Figure 6, which is difficult to see with the naked eye.
  9. The author needs to redraw, and none of the pictures are sharp enough.
  10. Section 4. The conclusions are too simplistic, I see no useful information in the conclusions, and the limitations of the research and the direction of future focus are not mentioned.

Author Response

The abstract is too long, and the abstract only needs to reflect the important results and conclusions of the study. Too many descriptive results in the abstract.
Answer: We checked and revised the abstracts according to your suggestion.

In the introduction, the author only introduces Korean radishes, which is one-sided because radishes are grown and processed in many countries around the world. What's the difference from the others? What are the advantages?
Answer: We described quality properties of radishes among some countries in lines 37-40. 

What are the differences between different varieties, even different colors and places of origin, and the components analyzed by LC-MS? The author should go into detail in the introduction.
Answer: we described simply them in lines 57-60 of the revised manuscript according to your comment.

The innovation of the research is not introduced and mentioned in the introduction.
Answer: We added the research meaning in lines 80-85 of the revised manuscript.

Among the materials, the radish is from December 2022 to February 2023. And the current opinion over the past two years, whether the processed materials still meet the food safety? Authors should provide data, such as microbes.
Answer: Purple radishes were planted in December 2022 in green house and harvested in February 2023. Purple radishes grown during about 90 days were immediately used in pickle processing. Therefore, we thought that the processed fresh materials was safety. To avoid confusion for the reader, we explained further information of purple radishes in lines 94-95 of the revised manuscript. We would be very grateful if you agree with us.

Fig.1 and Fig. 2 Because the combined picture is too small, it is difficult to see the content in the picture. Please modify and enlarge the picture and content.
Answer: The modified figures 1 and 2 were expressed in line 261 and line 318. The origin of figures is attached as PowerPoint and TIFF file. 

Section 3.5. In this part, the author only describes the results and phenomena, and the discussion on the results is insufficient.
Answer: We appreciated for your comment. We checked and revised Section 3.5 according to your suggestion.

The author should improve the clarity of Figure 6, which is difficult to see with the naked eye.
Answer: The modified Figure 6 was provided in line 429. The origin of Figure is attached as PowerPoint and TIFF file. 

The author needs to redraw, and none of the pictures are sharp enough.
Answer: We appreciated for your comment. We modified all Figures and attached their origins as TIFF file.

Section 4. The conclusions are too simplistic, I see no useful information in the conclusions, and the limitations of the research and the direction of future focus are not mentioned.
Answer: We revised the conclusion according to your suggestion.

Round 2

Reviewer 2 Report

Comments and Suggestions for Authors

Dear Editor and Authors,

Thank you for the invitation to review again (after revisions) the manuscript by Seung-Hun Chae , Sang-Hyeon Lee , Seung-Hwan Kim , Si-Hun Song , Jae-Hak Moon , Heon-Woong Kim , Jeong-Yong Cho: Changes in Quality and Metabolites of Pickled Purple Radish during Storage. In the reviewer's opinion, in this form it is suitable for publication.

Reviewer 3 Report

Comments and Suggestions for Authors

None.